# Upregulation of CD1d and ULBP3 on B cells from healthy donors and chronic lymphocytic leukaemia patients does not prime them for killing by γδ T cells

Julie David[1,2], Amy Walsh[1], Ke Sin Seow[1], Ellen Walsh[1,3], Stefan Elekes[1], Rohit Upadhyay[1], Nawal A. B. Taher[1¤], Samiha Al Siyabi[1], Ashanty M. Melo[1], Carmel Waldron[2], Elisabeth Vandenberghe[2], Anthony M. McElligott[2], Derek G. Doherty[1]*

1 Department of Immunology, School of Medicine, Trinity Translational Medicine Institute, St. James's Hospital, Dublin, Ireland, 2 Department of Haematology, School of Medicine, Trinity Translational Medicine Institute, St. James's Hospital, Dublin, Ireland, 3 Department of Genitourinary Medicine and Infectious diseases, St. James's Hospital, Dublin, Ireland

¤ Current address: Pharmacology Department, School of Medicine, Gharian University, Gharyan City, Libya

* derek.doherty@tcd.ie

## Abstract

Despite advances in treatment, chronic lymphocytic leukaemia (CLL) remains an incurable disease. Vδ1+ γδ T cells are reported to expand in CLL patients and to kill leukemic cells, placing them as candidates for immunotherapy. However, their cytotoxic efficacy was limited and required specific stimulatory conditions. Since some γδ T cells recognise lipids presented by CD1d, we examined if inducing CD1d expression and presentation of CD1d-restricted lipids could promote Vδ1, Vδ2 or Vδ3 T cell killing of B cells from CLL patients and healthy donors. Lines of γδ T cells containing Vδ1, Vδ2 and Vδ3 T cells and enriched cultures of CD19+ B cells were generated from peripheral blood using magnetic bead separation. γδ T cell subset frequencies and CD1d and ULBP3 expression levels on B cells were determined using flow cytometry. CD1d expression was induced on B cells by treatment with all trans retinoic acid (ATRA) and its analogue AM580. ATRA-treated B cells were co-cultured with γδ T cell lines in the absence or presence of lipids and cytotoxicity was assessed by measuring CD107a externalisation or propidium iodide staining using flow cytometry. Vδ1 and Vδ3 T cell frequencies were significantly higher in CLL patients compared to age-matched healthy donors. CD1d and ULBP3 expression was lower on CLL cells compared to healthy B cells but was restored by treatment with ATRA or AM580. Although co-culturing CLL cells with γδ T cells led to decreased cell viability, CD1d and ULBP3 upregulation by healthy and CLL B cells did not elicit cytolytic degranulation or cytokine production by Vδ1, Vδ2 or Vδ3 T cells, even after presentation of CD1d-restricted lipid antigens. This study suggests that human Vδ1 T

**Data availability statement:** All relevant data are within the paper and its Supporting Information files.

**Funding:** The author(s) received no specific funding for this work.

**Competing interests:** The authors have declared that no competing interests exist.

**Abbreviations:** α-GalCer, α-galactosylceramine; ATRA, all trans retinoic acid; CAR, chimeric antigen receptor; CLL, chronic lymphocytic leukaemia; Con A, concanavalin A; DCS, dead cell stain; E:T, Effector:target; FISH, fluorescent in situ hybridisation; FBS, foetal bovine serum; GDTC, γδ T cell; IBTS, Irish Blood Transfusion Service; IFN-γ, interferon-γ; IGHV, immunoglobulin heavy chain variable gene; iNKT, invariant natural killer T cell; LEF-1, lymphoid enhancer binding factor-1; MHC, major histocompatibility complex; MFI, mean fluorescence intensity; mAb, monoclonal antibody; NCR, natural cytotoxicity receptor; PBMC, peripheral blood mononuclear cell; PBS, phosphate buffered saline; PHA, phytohemagglutinin; RAR, retinoic acid receptor; TCR, T cell receptor; TMC, tetramyristoyl cardiolipin; TNF-α, tumour necrosis factor-α; ULBP, UL16-binding protein.

cell cytotoxicity against CLL B cells may be disease stage-dependent and requires B cell priming and selective activation of specific γδ T cell subsets.

## Introduction

Chronic lymphocytic leukaemia (CLL) is a B cell malignancy characterized by the expansion of a mature population of CD23+ CD5+ B cells [1]. The five-year survival rate is highly variable ranging from 20% to 90%, depending on genetic factors and disease risk status [2]. CLL has an incidence rate of approximately 4.7 per 100,000 people, making it the most common haematological malignancy in the USA and Europe, with Ireland having one of the highest incidence rates [3,4]. Although treatment for CLL has evolved significantly in recent years, with chemotherapy being replaced by novel targeted agents including the Bruton-tyrosine kinase inhibitor ibrutinib and the B-cell lymphoma-2 antagonist venetoclax [5–8], CLL remains largely incurable [9]. The highly heterogenous nature of the disease, the development of drug-resistant subclones, and the toxicity of current therapeutic regimes illuminates the need for immune-based therapies.

γδ T cells show promise as cellular therapies for CLL. γδ T cells are innate T cells that recognise non-peptide antigens without the need for major histocompatibility complex (MHC) presentation, and they contribute to immunity against tumours by direct killing and rapidly releasing cytokines that promote the activation of other anti-tumour effector cells [10,11]. γδ T cells comprise 1–5% of the entire T cell population in human blood and consist of 3 major subsets – Vδ1, Vδ2 and Vδ3 – based on their T cell receptor (TCR) δ-chain usage. Vδ2 T cells are the predominant subset in blood, while Vδ1 and Vδ3 T cells predominate in tissues, such as the gut, liver and adipose tissue [10–14]. The TCR of Vδ2 T cells recognises pyrophosphate intermediates presented by MHC class I-like antigen-presenting molecules, known as butyrophilins [14–15], whereas the Vδ1 and Vδ3 TCRs recognise a number of stress-inducible molecules expressed by virus-infected and tumour cells and lipid antigens presented by CD1c and CD1d molecules [16–19].

In addition to the TCR, all γδ T cell subsets express the activating receptor NKG2D, which can directly bind the stress-inducible molecules MIC-A and MIC-B and UL16-binding proteins (ULPBs), commonly upregulated on tumour cells [20–22]. Many γδ T cells also express natural cytotoxicity receptors (NCRs) which recognise ligands on tumour cells [12,14]. Upon stimulation through the TCR, NKG2D, NCRs, or cytokine receptors, all three subsets of γδ T cells display innate cytotoxic activity with the rapid production of Th1-type cytokines, such as interferon-γ (IFN-γ) and tumour necrosis factor-α (TNF-α), and the lytic molecules perforin and granzyme B [10–12,14].

A number of studies have revealed that the Vδ1 and Vδ3 subsets of γδ T cells are expanded in subgroups of patients with CLL [23–26]. Vδ1 T cells expanded from peripheral blood have been shown to display specific cytotoxicity towards B cells from CLL patients and CLL-derived cell lines *in vitro* [24,25,27], however, this cytotoxicity appears to be limited to particular properties of both the effector Vδ1 T cells

and the target CLL cells. Poggi et al. [24] found that CLL cells required activation with pokeweed mitogen to make them susceptible to killing by Vδ1 T cells and this killing was mediated by engagement of NKG2D by ULBP3. Correia et al. [27] found that Vδ1 T cells expanded under conditions that promoted the expression of NCRs, particularly NKp30, mediated direct cytotoxic killing of CLL cells isolated from patients. Vδ2 T cells from healthy donors but not from CLL patients were also demonstrated to kill CLL cells in vitro [28], however, other studies found that Vδ2 T cells did not kill CLL cells [24,27]. Vδ1 T cells have also been shown to inhibit tumour growth and to prevent tumour cell dissemination in xenograft models of CLL [29].

Since Vδ1, Vδ2 and Vδ3 T cells all can kill CLL cells under certain conditions, several groups have attempted to treat CLL cells to make them more susceptible to lysis by γδ T cells. Poggi et al. [24] treated CLL cells with the retinoic acid receptor-α (RAR-α) agonist all-trans retinoic acid (ATRA) to upregulate ULBP3 and MIC-A, which made them susceptible to killing by Vδ1 T cells. RAR-α agonists can also induce CD1c and CD1d expression on tonsillar B cells [30], and we reported that ATRA and its analogue AM580 can restore CD1d expression on CLL cells and healthy B cells, and these cells were capable of presenting glycolipid antigens to invariant natural killer T (iNKT) cells, resulting in cytotoxic killing of CLL cells [31]. CD1d is an MHC class I-like antigen-presenting molecule that presents phospholipid and glycolipid antigens to T cells [10,32]. CD1d is expressed across multiple B cell subsets in humans [33]. CD1d expression varies among B cell disorders [34,35] and is downregulated on malignant cells in CLL [31,34–37]. In addition to iNKT cells, Vδ1 and Vδ3 T cells can also recognise and respond to lipid antigens presented by CD1d and subsequently mediate a potent anti-tumour response [17–19].

The present study aimed to sensitise CLL cells for killing by γδ T cells through the upregulation of CD1d and ULBP3 on the surface of CLL cells and the presentation of CD1d-binding lipids to γδ T cells. By understanding the conditions under which γδ T cells can target CLL cells, we aim to evaluate the potential of Vδ1, Vδ2 and Vδ3 T cell-based immunotherapies for CLL, providing an alternative or adjunct to traditional treatments.

## Materials and methods

### Subjects and ethical statement

Anticoagulated blood samples were obtained from patients with CLL who presented to the Department of Haematology, St. James's Hospital, Dublin. This project was approved by St James's and Tallaght Hospitals Joint Research Ethics committee, and written informed consent was obtained from all participants. Healthy control samples were acquired either from consenting volunteers or from anticoagulated whole blood packs kindly provided by the Irish Blood Transfusion Service (IBTS) at St. James's Hospital. Peripheral blood mononuclear cells (PBMC) were prepared by standard density gradient centrifugation over Lymphoprep (STEMCELL Technologies) and cryopreserved in the Trinity/St. James's Hospital Haematology Biobank until needed.

### Antibodies and flow cytometry

PBMC or γδ T cell lines were stained with a dead cell stain (DCS; Fixable Viability Dye eFluor 506; eBiosciences) or propidium iodide (Miltenyi Biotec), human Fc blocking agent (Miltenyi Biotec) and the following mAbs: CD3 (clone HIT3A), CD19 (HIB19), CD1d (51.1), CD107a (H4A3), IFN-γ (4S.B3), TNF-α (Mab11) CTLA-4 (BNI3), TIM-3 (F38-2E2) and Vδ2 (B6) were purchased from Biolegend, CD5 (UCHT2) and Vδ1 (REA173) were purchased from Miltenyi Biotech; ULBP3 (166510) and LAG-3 (polyclonal IgG) were purchased from R&D Systems; PD-1 (EH12.1) was purchased from BD Pharmingen; and Vδ3 (P11.5B) was purchased from Coulter Immunotech. Following staining, the cells were acquired on a BD FACSCanto II flow cytometer (BD Biosciences) and analysed using FlowJo v10 software. The gating strategies used to measure CD1d and ULBP expression by B cells and CD107a expression by γδ T cell subsets are shown in S1 Fig. CLL cell viability was measured as uptake of propidium iodide (PI) by CD19$^+$/CD5$^+$ cells. Incubation with 0.18% (w/v) Triton-X 100 was used as a positive control of CLL cell death.

## Enrichment of peripheral B cells

Flow cytometry confirmed that B cells were highly enriched in the CLL patients but accounted for less than 10% of lymphocytes from healthy donors, therefore, B cells were enriched from healthy donor PBMC only, resulting in similar frequencies to those found in PBMC from CLL patients (S1C Fig). B cells were enriched from $2x10^8$ PBMC by positive magnetic bead selection using CD19 Microbeads, followed by selection using MACS MS Cell Separation column according to the manufacturer's instructions (Miltenyi Biotec). The enriched B cells were counted and plated in complete RPMI medium (RPMI 1640 with Glutamax containing 25 mM HEPES, 50 μg/ml streptomycin, 50 U/ml penicillin and 10% heat-inactivated foetal bovine serum (FBS)) in 96-well round bottom cell culture plates.

## Treatment of B cells with RAR-α agonists

B cells from healthy donors and CLL patients were treated for 24–72 hours with ATRA, AM580 or retinol (Sigma-Aldrich) at 30, 300 and 3000 ng/mL. Cells were cultured in a 5% $CO_2$ incubator at 37˚C and subsequently analysed for CD1d and ULBP3 expression by flow cytometry.

## Generation of primary cultures of γδ T cells

Primary cultures of γδ T cells, which contained Vδ1, Vδ2, and Vδ3 T cells, were generated from $2x10^8$ PBMC isolated from healthy donors. Following treatment with human FcR blocking reagent (Miltenyi Biotec), the PBMC were incubated with a CliniMACS TCR α/β-Biotin mAb followed by anti-biotin Microbeads, as per the supplier's instructions (Miltenyi Biotec) and the αβ TCR+ T cells were then depleted using a MACS LD Cell Separation column. The αβ T cell-depleted fraction was then resuspended at a density of $7.5x10^6$ cells/mL in 'γδ T cell medium' (RPMI Glutamax medium containing 10% FBS, 25 mM HEPES, 150 μg/ml streptomycin, 50 U/ml penicillin, 1 mM sodium pyruvate, 50 μM 2-mercaptoethanol, 1% non-essential amino acids and 1% essential amino acids) and stimulated with 1 μg/mL anti-CD3 antibody (clone OKT3, BioLegend). Cells were cultured in 96-well round-bottom plates at 37˚C and 5% $CO_2$. After 24 hours, and every 4–5 days thereafter, the medium was replaced with fresh medium containing 100 U/mL IL-2 and 70 ng/mL of IL-15 (Miltenyi Biotec). Cells were cultured for up to 60 days and the purities and relative frequencies and phenotypes of Vδ1, Vδ2 and Vδ3 T cells were assessed by flow cytometry. Ten lines of γδ T cells were used in the present study. At the time of use in experiments, Vδ1 T cells accounted for 13.9–51% (mean 29.2%) of the cells, whereas Vδ2 T cells accounted for 18.5–82.0% (mean 45.7%) and Vδ3 T cells accounted for 1.3–25.0% (mean 8.3%). The remaining cells were Vδ1- Vδ2- Vδ3- γδ T cells and no αβ T cells, B cells or NK cells were present.

## Analysis of cytolytic degranulation by γδ T cells

Cytotoxic responses of γδ T cells were recorded through flow cytometric measurement of cell-surface CD107a expression, a marker of cellular degranulation. CD19+ B cells from healthy donors or PBMC from CLL patients and healthy controls, were plated in 96-well, round bottom microtitre plates at densities of $0.25x10^6$ cells per well and treated for 24–72 hours with various concentrations of RAR-α agonists, as described above. Equal numbers of expanded γδ T cells were added. Phorbol myristate acetate (50 ng/mL; Sigma-Aldrich) and ionomycin (1 μg/ml; Sigma Aldrich) or anti-CD3 and anti-CD28 mAbs (1 μg/mL each) were added to positive control wells containing γδ T cells alone. Fluorochrome labelled anti-CD107a antibody was added to each well. After 1 hr, monensin (BioLegend) at a final concentration of 5 nmol/well was added and left to incubate for a further 3 hr. The cells were then stained for analysis by flow cytometry.

## Priming CLL cells for cytotoxicity by γδ T cells using glycolipids

B cells were also treated with 5 glycolipids, that have been shown by us (S2 Fig, ref [38]) and others [17,39–41] to bind to CD1d and activate T cells. The structures of these glycolipids are shown in Fig 1. Sulfatide, lyso-sulfatide, cardiolipin and tetramyristoyl cardiolipin (TMC) were purchased from Avanti Polar Lipids and α-galactosylceramide (GC) was obtained

**Fig 1. Glycolipids used in the present study.** Sulfatide derives from galactosylceramide (GalCer) via esterification of a sulphate group to 3-hydroxyl of the galactose moiety. Lyso-sulfatide lacks the fatty acid chain of sulfatide. Cardiolipin consists of 2 phosphate residues and 4 fatty acyl chains. Tetramyristoyl cardiolipin (TMC) is a synthetic analogue of cardiolipin.

from Funakoshi. The glycolipids were prepared in dimethylsulphoxide (DMSO) to a stock concentration of 1 mM and stored at -20°C. Following thawing at room temperature, glycolipids were vortexed for 1 minute, heated at 80°C for 2 minutes and sonicated for 10 minutes before diluting them in medium and adding them to the B cells or PBMC at 100 ng/mL. After incubating the glycolipid-treated cells for 16 hours, equal numbers of expanded γδ T cells were then added into each well and the CD107a degranulation assay was carried out as described above.

## Statistical analysis

Statistical analysis was carried out using GraphPad PRISM 8.0.2 for Windows. Mann-Whitney U test was used to compare unpaired, independent samples. Wilcoxon signed-rank test was used to compare paired, dependent samples. One-way ANOVA and post hoc Dunnett's multiple comparison test were used to compare the means of three or more groups. Repeated measures Two-Way ANOVA and post hoc Tukey's test was used to compare group means across matched samples. Statistical significance was reported as $p < 0.05$.

## Results

### Characteristics of the CLL patients

Eighteen patients with CLL were recruited for this study. The patient demographics are shown in Table 1. Of the total patient population, 72% were male. The participant's age ranged from 36–83 years (median, IQR 24), with 44% of the

cohort being ≥ 65 years. Of the sixteen tested, three individuals (18.75%) in the cohort tested positive for the expression of CD38, a well-classified negative prognosticator. Of the twelve individuals in which CD49 expression was tested (Patients 6–15, 17–18 in Table 1], six (50%) were positive. Of the nine individuals in which the immunoglobulin heavy chain variable (IGHV) gene region mutational status was determined (Patients 1, 2, 3, 6, 7, 10, 11, 14 and 15 in Table 1], four patients (~44%) and five patients (~55%) had IGHV-mutated and -unmutated disease, respectively. Of the thirteen individuals in which TP53 mutational status was determined (Patients 1, 2, 3, 6, 7, 9, 10, 11, 12, 14, 15, 17 and 18 in Table 1], one patient (~8%) had TP53 mutated disease. Chromosomal abnormalities were measured in eleven patients by FISH (fluorescent in situ hybridisation), in which 36.7% carried the Del13q aberration. One case of trisomy 12 (0.9%) (Patient 5) and another of Del11q (0.9%) (Patient 11) was detected, with no aberrations found in 45% of patients. Approximately 60% of patients received treatment prior to sample biobanking.

Sixteen age-matched healthy donors were recruited for phenotypic comparisons with the CLL patients. Blood samples from a further 27 anonymous healthy donors were obtained for cell enrichment and functional studies.

### Reduced CD1d expression by B cells and increased Vδ1 and Vδ3 T cell frequencies in CLL patients

PBMC were isolated from 16 healthy donors and 13 patients with CLL and stained with mAbs specific for CD3, CD19, Vδ1, Vδ2 and Vδ3 and analysed by flow cytometry (S1 Fig). Fig 2 shows that the frequency of Vδ1 and Vδ3 T cells, as percentages of total T cells, was significantly increased in CLL patients compared to age-matched healthy donors (Fig 2A,C). The frequencies of Vδ2 T cells were similar in CLL patients and age-matched healthy controls (Fig 2B). As expected, B cells were significantly expanded within PBMC of the CLL patients (Fig 2D). Phenotypic analysis of B cells

**Table 1. Patient demographics and disease characteristics.**

| Patient | Sex | Age | Flow cytometry | IGHV/TP53 status | FISH | Pre-treated? | Treatment[1] |
|---|---|---|---|---|---|---|---|
| 1 | M | 78 | CD38+ | UM/UM | No aberrations | Y | Retuximab, cyclophosphamide x6 |
| 2 | F | 54 | CD38- | MU/UM | Del 13q | N | n/a |
| 3 | M | 59 | NT | MU/UM | Del 13q | Y | Retuximab-bendamustine |
| 4 | F | 83 | CD38- | NT | NT | Y | Chlorambucil |
| 5 | M | 80 | CD38- | NT | Trisomy 12 | N | n/a |
| 6 | M | 54 | CD38- CD49d+ | UM/UM | No aberrations | Y | FCR, retuximab-bendamustine, ibrutinab |
| 7 | F | 57 | CD38- CD49d- | MU/UM | Del 13q | N | n/a |
| 8 | F | 71 | CD38- CD49d+ | NT | No aberrations | Y | FCR x4, retuximab-bendamustine x4, ibrutinib |
| 9 | M | 74 | CD38- CD49d+ | NT/MU | NT | Y | FCR x6, zanubrutinib |
| 10 | M | 49 | CD38- CD49d+ | UM/UM | No aberrations | N | n/a |
| 11 | M | 55 | CD38- CD49d- | UM/UM | Del 11q | N | n/a |
| 12 | M | 83 | CD38 + CD49d- | NT/UM | NT | Y | FCR x6, IV-Ig, azathioprine + retuximab x4 |
| 13 | M | 73 | CD38- CD49d+ | NT | Del 13q | N | n/a |
| 14 | M | 51 | CD38- CD49d- | MU/UM | No aberrations | Y | Venetoclax + obinutuzumab |
| 15 | M | 36 | CD38 + CD49d- | UM/UM | NT | Y | FCR x4, ibrutinib |
| 16 | F | 74 | NT | NT | NT | N | n/a |
| 17 | M | 68 | CD38- CD49d- | NT/UM | NT | N | n/a |
| 18 | M | 48 | CD38- CD49d+ | UM/UM | NT | Y | Venetoclax + obinutuzumab |

F, female; FCR, fludarabine, cyclophosphamide and rituximab; FISH, fluorescent in situ hybridisation; IGHV, immunoglobulin heavy chain; IV-Ig, intravenous immunoglobulin; M, male; MU, mutated; N, no; n/a, not applicable; NT, not tested; UM, unmutated; Y, yes.

[1]No patients were on treatment at the time of study.

[2]Patient 12 underwent Richter transformation from underlying CLL to an anaplastic large cell lymphoma requiring new treatment with each relapse. Patient 12 also developed immune thrombocytopenia.

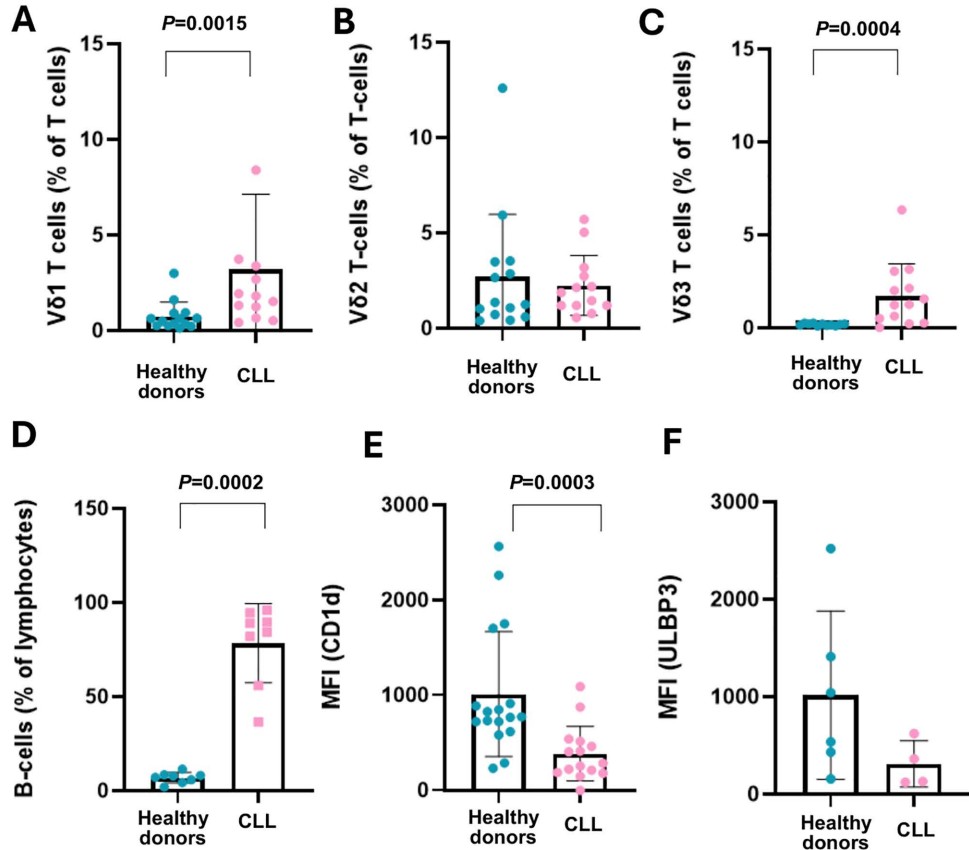

**Fig 2. Phenotypic analysis of PBMCs from CLL patients and healthy donors.** PBMC were isolated from healthy donors and patients with CLL. PBMC from healthy donors were enriched for CD19+ B-cells using magnetic bead separation. Cells were stained with mAbs specific for CD3, CD19, Vδ1, Vδ2, Vδ3, CD1d and ULBP3 and analysed by flow cytometry. Scatter bar plots depict the frequencies of T cells from healthy donors and CLL patients that expressed Vδ1 **(A)**, Vδ2 **(B)** and Vδ3 **(C)** TCRs, the percentages of lymphocytes that are B cells **(D)**, and the mean fluorescence intensities of CD1d **(E)** and ULBP3 **(F)** expression on CD19+ B-cells. The data in **(A)**, **(B)** and **(C)**, show means ± standard deviation (SD) of 13 CLL patients and 11-14 age-matched control subjects. The data in **(D)** show means ± SD of 8 CLL patients and 8 healthy donors. The data in **(E)** show means ± SD of 14 CLL patients and 16 control subjects. The data in **(F)** show means ± SD of 6 healthy subjects and 4 CLL patients. Statistical significance was determined using the non-parametric Mann-Whitney U test and statistical differences are indicated.

revealed that B cells from CLL patients expressed significantly lower levels of CD1d, as measured by median fluorescence intensity (MFI) (Fig 2E). ULBP3 expression by B cells was also lower in CLL patients compared to control subjects, but this difference was not significant (Fig 2F).

**γδ T cells do not spontaneously degranulate in response to B cells from CLL patients or healthy donors**

We next investigated if γδ T cells from healthy donors could kill B cells from CLL patients and healthy donors. Lines of γδ T cells were generated from 5 healthy donors under conditions that promote the growth of Vδ1, Vδ2 and Vδ3 T cells (S1B Fig). The γδ T cell lines were co-cultured with equal numbers of PBMC from 9 CLL patients and B cells enriched from PBMC from 10 healthy donors in the presence of an anti-CD107a mAb. Cells were then stained with mAbs specific for CD3, CD19, Vδ1, Vδ2 and Vδ3 and the expression of the degranulation marker CD107a by individual Vδ1, Vδ2 and Vδ3 T cells was analysed by flow cytometry. As seen in Fig 3A, no γδ T cell subset displayed increased degranulation in response to B cells from CLL patients or healthy donors. The expression of inhibitory molecules on γδ T cell lines at Day

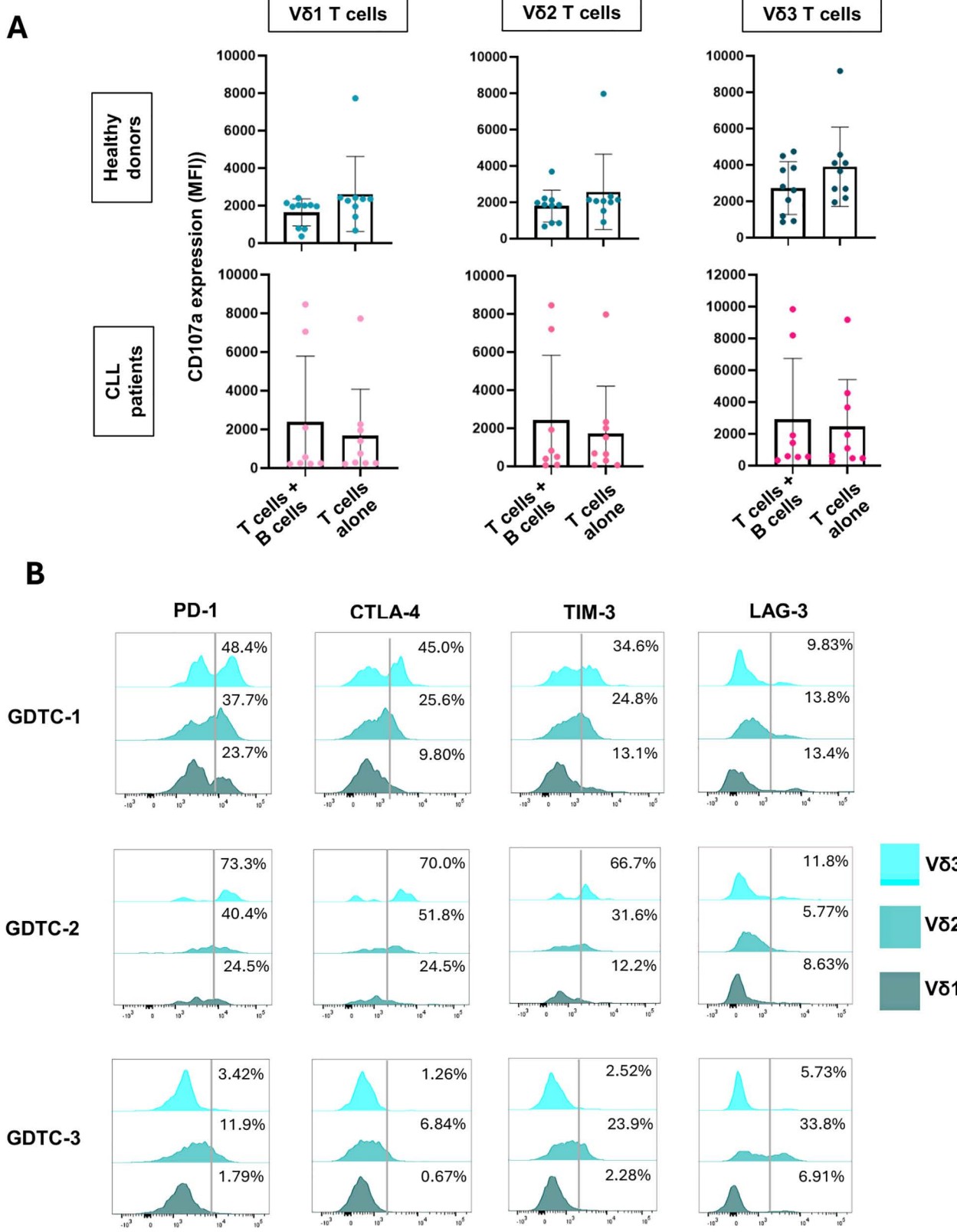

**Fig 3. γδ T-cells do not degranulate in response to B cells from CLL patients or healthy donors.** PBMC from 9 CLL patients and B cells enriched from PBMC from 10 healthy donors were co-cultured with lines of expanded γδ T cell lines from healthy donors in the presence of an anti-CD107a mAb.

Cells were then stained with mAbs specific for CD3, CD19, Vδ1, Vδ2 and Vδ3 and analysed by flow cytometry. **(A)** Scatter bar plots depict the mean fluorescence intensities (MFI) of CD107a expression by gated Vδ1 (left panels), Vδ2 (centre panels) and Vδ3 (right panels) T cells cultured alone or with B cells from healthy donors (n = 10) (upper panels) and CLL patients (n = 9) (lower panels). These data show means ± standard deviation. MFIs of CD107a expression by γδ T cell subsets cultured alone were compared to when cultured with B cells using the using the non-parametric Wilcoxon matched pairs signed rank test and no significant differences were found. **(B)** Flow cytometry analysis of inhibitory receptor expression by γδ T cell lines (rows) and subsets (columns) at Day 24 in culture. Cells were stained with mAbs specific for CD3, CD19, Vδ1, Vδ2, Vδ3, PD-1, CTLA-4, TIM-3 and LAG-3 and analysed by flow cytometry.

24 was evaluated, directly before their use in cytotoxicity assays. As seen in Fig 3B, High expression of PD-1 and CTLA-4 was observed on Vδ2 and Vδ3 T cells from two γδ T cell lines (GDTC 1, GDTC 2). Moderate expression of TIM-3 was observed on Vδ2 and Vδ3 T cells from two γδ T cell lines (GDTC 1, GDTC 2), while LAG-3 was lowly expressed on all three lines. These results suggest that γδ T cells do not spontaneously degranulate in response to CLL cells or healthy B cells under standard culture conditions, and this may be partially due to functional exhaustion of long-term expanded γδ T cell lines.

### The viability of CLL cells decreases in co-culture with γδ T cells and this is not enhanced by ATRA

The viability of CLL cells in co-culture with γδ T cells was next assessed at varying E:T ratios. Lines of γδ T cells were generated from 3 healthy donors under conditions that promote the growth of Vδ1, Vδ2 and Vδ3 T cells (S1B Fig). *Ex vivo* CLL PBMC were plated and co-cultured with equal numbers (1:1), three-fold (3:1) or six-fold (6:1) as many γδ T cells for 4 hr. Cells were then harvested, stained with propidium iodide and mAbs specific for CD19 and analysed by flow cytometry. Contrasting with previous results, the % dead CLL cells increased upon co-culture with expanded γδ T cell lines, and this became more pronounced at higher E:T ratios (Fig 4.). The previous lack of γδ-mediated cytotoxic degranulation suggests that this drop in viability may reflect either non-degranulating cytotoxic mechanisms, cytotoxicity mediated by γδ T cells other than Vδ1, Vδ2 and Vδ3 T cells, or may be due to highly stressful co-culture conditions.

### Upregulation of CD1d and ULBP3 on B cells from CLL patients and healthy controls by ATRA does not sensitise them for killing by γδ T cells

We hypothesised that treatment of B cells from CLL patients with RAR-α agonists would induce upregulation of CD1d and ULBP3, which would provide stimulatory signals for TCR- and NKG2D-mediated γδ T cell activation, respectively, resulting in B cell killing [24,31]. PBMC from 5 CLL patients and B cells enriched from PBMC from 10–13 healthy donors were treated for 72 hr with 30, 300 or 3,000 ng/mL ATRA, AM580 and retinol. The cells were then stained with mAbs specific for CD19, CD1d and ULBP3 and analysed by flow cytometry. Fig 5A,C shows that treatment with ATRA and AM580 led to a significant increase in CD1d expression on both healthy donor B and CLL B cells, whereas retinol had no effect on the expression of either marker. Upregulation of CD1d and ULBP3 by ATRA appeared to follow a dose dependence, while upregulation by AM580 did not. Fig 5B shows that treatment with ATRA also led to a significant increase ULBP3 expression on healthy B cells. This upregulation was not achieved by AM580 or retinol stimulation.

We next investigated if treatment of B cells with RAR-α agonists would make them susceptible to killing by Vδ1, Vδ2 or Vδ3 T cells through upregulation of CD1d and ULBP3 on their cell surface. Enriched B cells from five healthy donors and PBMCs from five CLL patients were treated for 72 hr with 30 ng/mL RAR-α agonists. The cells were then co-cultured with equal numbers of expanded γδ T cell lines from healthy donors in the presence of an anti-CD107a mAb. Cells were then stained with mAbs specific for CD3, CD19, Vδ1, Vδ2 and Vδ3 and analysed by flow cytometry. Fig 6 A shows that treatment of B cells from CLL patients or healthy donors with any of the RAR-α agonists at 30 ng/mL did not trigger cytolytic degranulation by Vδ1, Vδ2 or Vδ3 T cells. Thus, although ATRA and AM580 can induce the expression of stimulatory

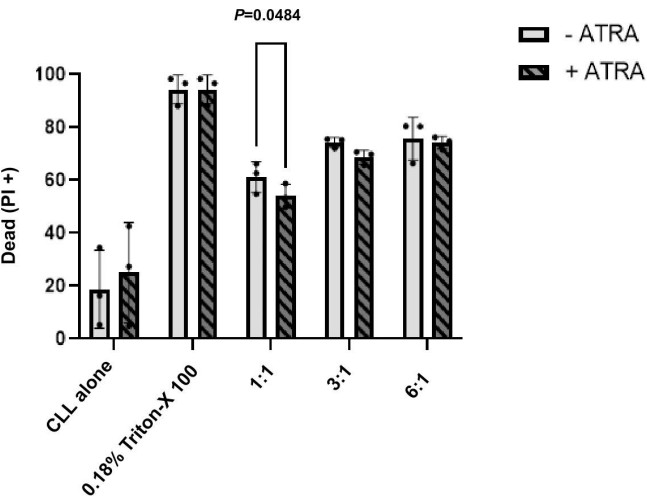

**Fig 4. Dose dependent reduction in CLL cell viability inγδ T cell co-cultures.** PBMC from 3 CLL patients were treated for 72 hr with medium alone or 30 ng/mL ATRA. The cells were then co-cultured with expanded γδ T cell lines from healthy donors for 4 hr (n = 3). Viability of CLL cells was measured as PI uptake by CD19+/CD5+ cells by flow cytometry. Scatter bar plots compare the % dead (± SD) (PI+) CLL cells with and without treatment with ATRA. Data were compared using repeated measures Two-way ANOVA and post hoc Tukey's multiple comparison test and significant differences (P < 0.05) are indicated.

ligands on B cells, this did not lead to cytolytic degranulation by any of the subsets of γδ T cells. The production of two key pro-inflammatory cytokines, IFN-γ and TNF-α, by γδ T cell subsets was next assessed in response to CLL cells treated with or without ATRA. Although Vδ1, Vδ2 and Vδ3 T cells produced both cytokines in response to PMA and ionomycin stimulation, no difference in IFN-γ or TNF-α production, and therefore no non-cytolytic functionality, was observed in co-culture conditions with or without pre-treatment of ATRA.

### Addition of CD1d-binding glycolipids to B cells from healthy donors and CLL patients does not sensitise them for killing by Vδ1, Vδ2 and Vδ3 T cells

It was hypothesised that the addition of CD1d-binding glycolipids may sensitise B-cells from healthy donors and CLL patients to TCR-mediated killing by Vδ1, Vδ2 and Vδ3 γδ T-cells. Five different glycolipids; α-GC, sulfatide, lyso-sulfatide, cardiolipin and TMC were vortexed, heated, sonicated and added to enriched B-cells from healthy donors or CLL cells +/-ATRA and left overnight. The cells were then co-cultured with equal numbers of expanded γδ T cell lines generated from healthy donors in the presence of anti-CD107a mAb. Cells were then stained with mAbs specific for cell surface expression of CD3, CD19, Vδ1, Vδ2 and Vδ3 T cells and analysed by flow cytometry. Fig 7 shows that the addition of glycolipid antigens does not induce killing of healthy donor B cells or CLL cells by any γδ T-cell subset.

## Discussion

γδ T cells have emerged as promising candidates for cancer immunotherapy due to their ability to recognize a broad range of tumour-associated ligands, their potent cytotoxicity, efficient tumour infiltration, and their capacity to stimulate adaptive immune responses [10–14,42]. Unlike conventional CD8+ T cells, γδ T cells are not MHC-restricted, meaning they can potentially be used as allogeneic 'off-the-shelf' therapies for multiple patients without inducing graft-versus-host disease. Multiple clinical trials are currently exploring the therapeutic potential of γδ T cells – including those expressing chimeric antigen receptors (CAR-γδ T cells) – for various cancers, including CLL [11,43].

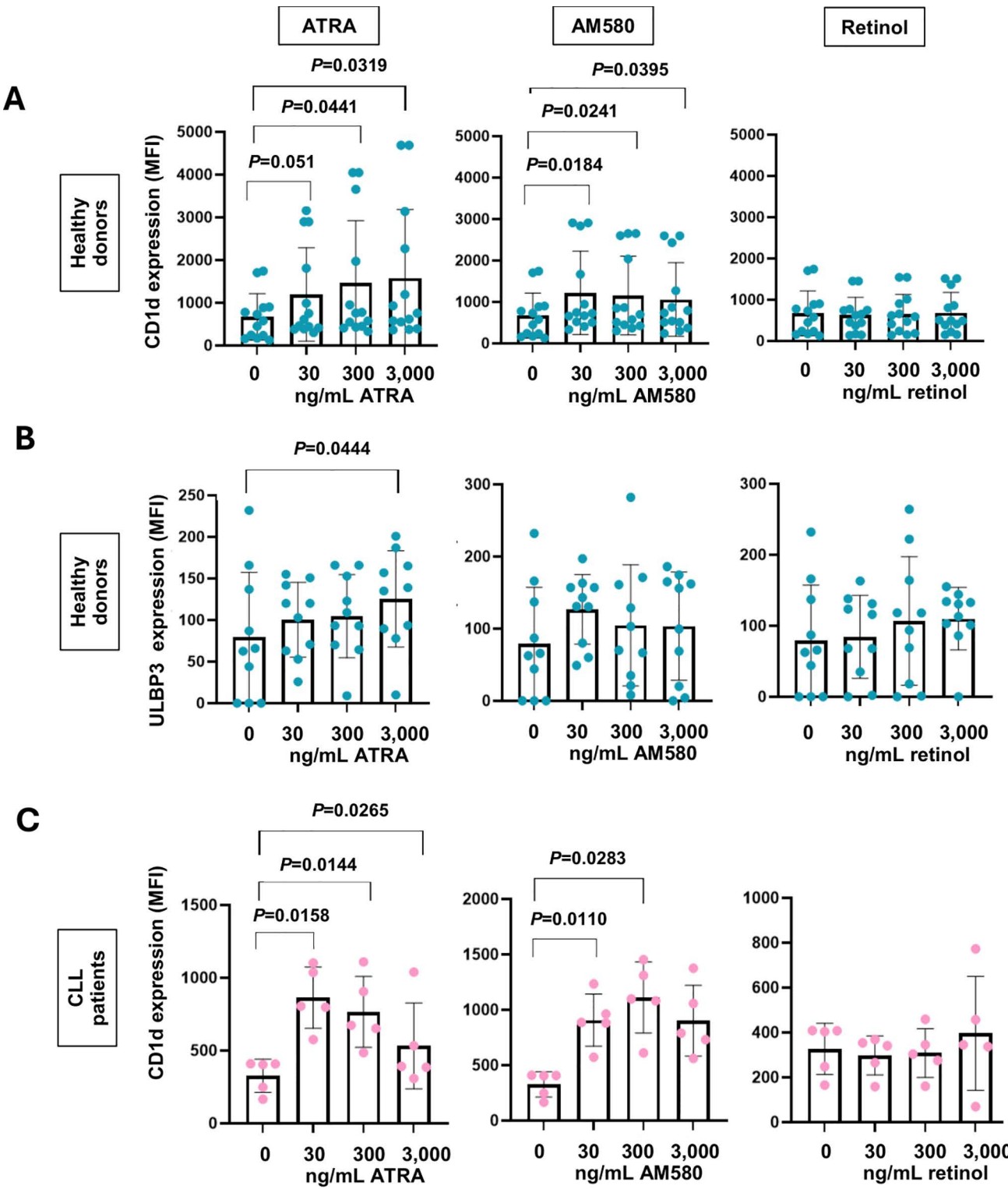

**Fig 5. Upregulation of CD1d and ULBP3 on B cells from healthy controls and CLL patients using RAR-α agonists.** PBMC from 5 CLL patients and B cells enriched from PBMC from 10-13 healthy donors were treated for 72-96 hr with ATRA, AM580 and retinol at the indicated concentrations. The cells were subsequently stained with mAbs specific for CD1d or ULBP3 and analysed by flow cytometry. Scatter bar plots depict the geometric median

fluorescence intensities (± SD) of CD1d (n = 13) (**A**) and ULBP3 (n = 10) (**B**) expression on B cells from healthy donors following treatment with 30, 300 and 3000 ng/ml ATRA (left panels), AM580 (centre panels) and retinol (right panels). (C), Scatter bar plots depict the geometric MFI (± SD) of CD1d expression on B cells from CLL patients. Data were compared using One-way ANOVA and post hoc Dunnett's multiple comparisons test and significant differences (P < 0.05) are indicated.

A number of studies have reported expansions of Vδ1, Vδ2, and/or Vδ3 T cells in patients with CLL [23–26]. These cells can be induced to kill CLL B cells under certain stimulatory conditions [24,25,27], making them candidates for cell-based therapy. However, the cytotoxic efficacy of these cells in CLL was limited and required specific stimulatory conditions. In the present study, we found that the frequencies of both peripheral blood Vδ1 and Vδ3 T cells, as percentages of total T cells, were significantly increased in CLL patients compared to those in age-matched healthy donors. Two patients with CLL demonstrated abnormally high levels of Vδ1 T cells (8.3 and 14.2%, respectively), one of which harboured a trisomy 12 chromosomal aberration. This finding may support previous statements describing a correlation between disease severity and Vδ1 T cell frequencies. Vδ2 T cell frequencies were similar in both groups. Importantly, when γδ T cells (including Vδ1, Vδ2, and Vδ3 subsets) were co-cultured with CLL B cells, there was no observable cytolytic activity, as indicated by a lack of CD107a externalisation, or pro-inflammatory cytokine production. The expansion of Vδ1 and Vδ3 T cells observed in this cohort is consistent with previous reports, however, these cells lack intrinsic cytotoxic activity against CLL cells, as has also been described [24,25,27]. As discussed below, this discrepancy may stem from differences in patient populations, disease stages or methodological approaches.

To investigate why γδ T cells failed to recognize or kill CLL cells, the expression of two key ligands - CD1d and ULBP3 – on B cells was analysed. Confirming previous reports [31,34–37], we found that both ligands are expressed at markedly lower levels on B cells from CLL patients compared to B cells from healthy donors. CD1d can present glycolipid antigens to the TCRs of Vδ1 and Vδ3 T cells [10,17–19], whereas ULBP3 binds to NKG2D, a stimulatory receptor found on all γδ T cell subsets [20,21]. The observed downregulation may represent an immune evasion strategy used by CLL cells, to limit the antigenic exposure of CLL cells to T cells, similar to the downregulation of MHC class I seen in other cancers [44]. One mechanism might be the overexpression of lymphoid enhancer-binding factor-1 (LEF-1), a transcriptional repressor of CD1d known to be upregulated in CLL patients [45].

We attempted to prime CLL cells for killing by treating them with ATRA and its analogue AM580, which have been shown by us and others to induce both CD1d and ULBP3 expression on B cells. ATRA has been used clinically to sensitize CLL cells to fludarabine-induced apoptosis [46] and is a standard therapy in acute promyelocytic leukaemia [47]. Consistent with prior findings [24,30,31], we found that treatment with ATRA and AM580 (but not retinol) significantly increased CD1d expression on both healthy and CLL B cells. Additionally, treatment with 3000 ng/mL ATRA resulted in significant upregulation of ULBP3 on healthy donor B cells. However, despite upregulation of these ligands, γδ T cells still failed to degranulate in response to treated CLL cells, suggesting that ligand expression alone is not sufficient to induce cytotoxicity. Interestingly, the viability of CLL cells was seen to decrease upon co-culture with γδ T cells, an observation which was more pronounced at greater E:T ratios. This drop in viability may reflect either non-degranulating cytotoxic mechanisms, such as those associated with death receptor ligation, or cytotoxicity mediated by γδ T cells other than Vδ1, Vδ2 and Vδ3 T cells (which typically account for 1–5% of cells in the γδ T cell lines).

Alternatively, the drop in viability may be the result of the highly stressful co-culture conditions employed, which could reflect nutrient deficiency, local hypoxia, lactate accumulation or harsh conditions reflected in physically crowded co-culture conditions [48–50]. The substantial cell density needed for high E:T ratios and stressful culture conditions as a result may explain this loss of viability independent of γδ T-cell cytotoxicity. CLL cells are metabolically fragile and notoriously sensitive to oxidative stress and changes in pH and nutrient availability, all of which are heavily impacted by co-culture conditions with high E:T ratios. Given that ATRA significantly upregulates both CD1d and ULBP3 on CLL cells,

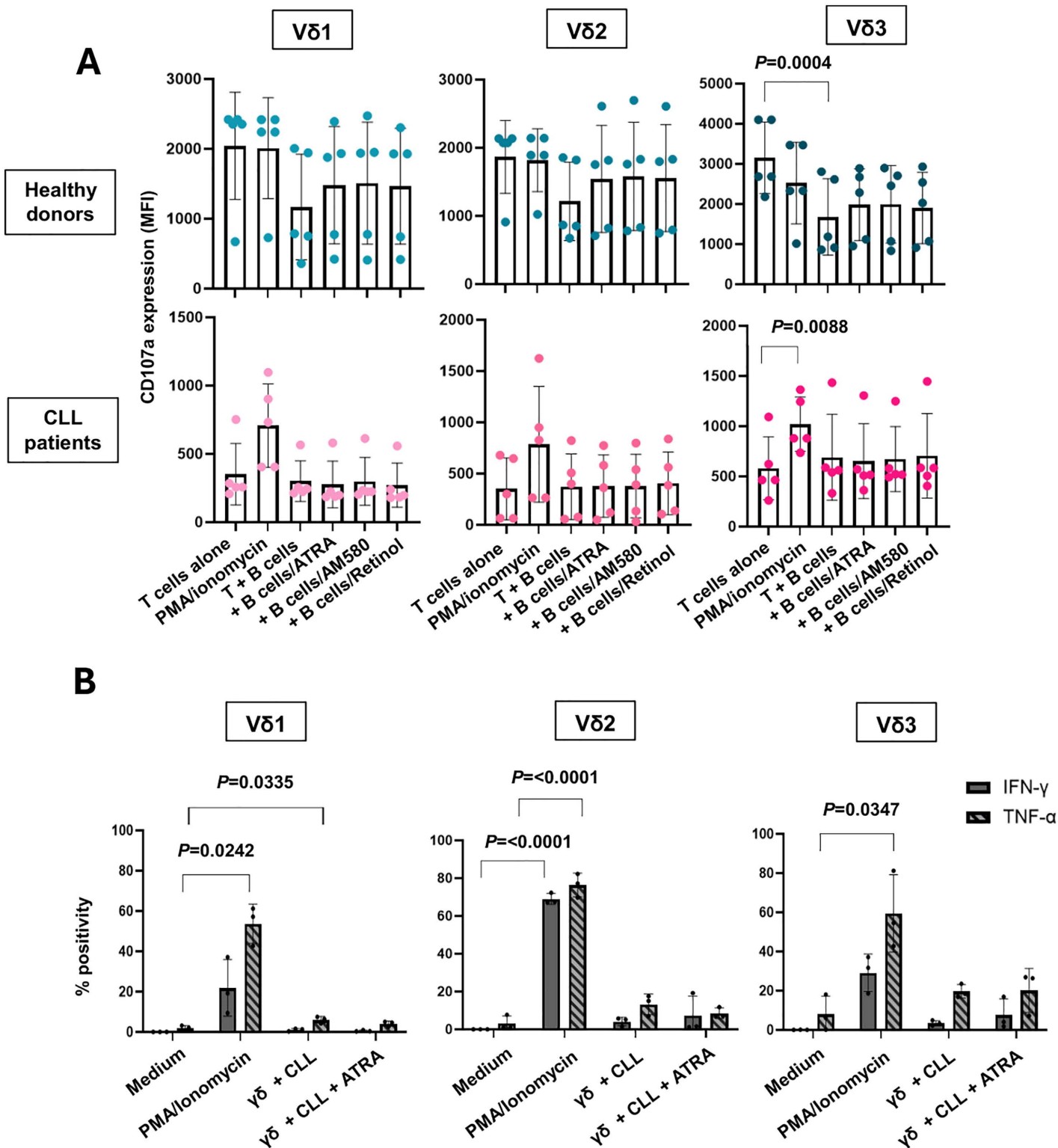

**Fig 6. RAR-α agonist treatment of B cells from CLL patients and healthy donors did not sensitise them for killing by γδ T-cells. (A)** Enriched B cells from 5 healthy donors and PBMCs from 5 CLL patients were treated for 72 hr with RAR-α agonists at 30 ng/mL. The cells were then co-cultured with expanded γδ T cell lines from healthy donors in the presence of an anti-CD107a mAb for 4 hr (n = 5). Cells were then stained with mAbs specific for CD3, CD19, Vδ1, Vδ2 and Vδ3 and analysed by flow cytometry. Scatter bar plots depict the mean (± SD) fluorescence intensities (MFI) of CD107a

expression by gated Vδ1 (left panels), Vδ2 (centre panels) and Vδ3 (right panels) T cells cultured alone or with the treated B cells. **(B)** PBMCs from 3 CLL patients were treated for 72 hr with medium alone or ATRA at 30 ng/mL. The cells were then co-cultured with expanded γδ T cell lines from 3 healthy donors for 4 hr. The γδ T cells were analysed for IFN-γ and TNF-α production by flow cytometry. Percent positivity of IFN-γ and TNF-α expression by Vδ1 (left panel), Vδ2 (centre) and Vδ3 (right) T cells was compared using a One-way or Two-way ANOVA and post hoc Tukey's multiple comparison test and significant differences (P < 0.05) are indicated.

and given the viability of CLL cells did not change between ATRA-treated and -untreated cells at any E:T ratio, this supports that upregulation of CD1d and ULBP3 on B cells from healthy donors and chronic lymphocytic leukaemia patients does not prime them for killing by γδ T cells.

CD1d-dependent killing requires the presentation of lipid or glycolipid antigens to T-cells [10,17,18,31–33]. It was originally hypothesised that CLL cells would produce an endogenous tumour-associated lipid antigen that would bind to CD1d and stimulate γδ T-cell cytotoxicity. Since this was not the case, and because iNKT cell agonist glycolipid α-galactosylceramide is required to prime CLL cells for killing by iNKT cells [31], we next tested five synthetic glycolipids for their ability to induce γδ T cell killing of untreated or ATRA-treated B cells. These glycolipids – α-galactosylceramide, sulfatide, lyso-sulfatide, cardiolipin, and TMC – have previously been shown to activate T cells, including γδ T cells, via CD1d presentation [17,38–41]. However, none of these glycolipids succeeded in priming B cells for killing by Vδ1, Vδ2 or Vδ3 T cells even after treatment with ATRA to upregulate CD1d expression.

Several methodological differences may explain the lack of cytotoxicity observed. Firstly, the variable patient characteristics may produce different findings. Our study cohort included eighteen CLL patients with variable clinical backgrounds and considerable diversity in disease severity and treatment history. Some had high-risk genetic abnormalities such as TP53 mutations or 11q deletions with unmutated IGHV genes [51,52]. Over half had received treatment prior to sample collection, including a patient with relapsed disease or Richter transformation. Chemotherapy-induced mutations and immune evasion mechanisms in these patients may have impacted the functionality and expression of ligands on CLL cells, thus influencing γδ T cell recognition and activity. Interestingly, Poggi et al. [24] reported that patients with high-risk CLL did not exhibit expansions of cytotoxic Vδ1 T cells and showed resistance to ULBP3 induction by ATRA. The high proportion of patients with adverse prognostic markers in our study may explain the lack of Vδ1 T cell cytotoxicity observed.

The failure of γδ T cells to kill ATRA-treated B cells may reflect differences in γδ T cell line and CLL cell treatments in different studies. Poggi et al. [24] reported that Vδ1 T cells from CLL patients were only capable of lysing CLL cells if first treated with pokeweed mitogen or 10 μg/mL ATRA to upregulate MIC-A or ULBP3. This concentration of ATRA is much higher than that used in our study (30 ng/mL). The current study used mixed polyclonal γδ T cell lines derived from healthy donors, contrasting with the CD4- CD8- individual Vδ1 and Vδ2 T cell lines generated by Poggi et al. [24]. Indeed, CD1d-dependent cytotoxicity mediated by iNKT cells against CLL cells was limited to CD8+ iNKT cells, whereas their CD4+ and CD4-CD8- counterparts did not exhibit cytotoxic activity [24] and the same might be true for Vδ1 T cells. Correia and co-workers [27] reported that purified NKp30+ Vδ1 T cell lines expanded with IL-2 and phytohemagglutinin, exhibited cytotoxicity against leukemic cell lines and patient-derived CLL cells, but NKp30- Vδ1 T cells were not cytotoxic. The expression of NKp30 was not investigated in the present study. Siegers et al. [25] expanded Vδ1 T cells from isolated pan-γδ T cell lines with concanavalin A for 6–8 days and these Vδ1 T cells were cytotoxic towards two CLL cell lines that do not express ULBP3 or MIC-A/B but expressed TRAILR1. These authors reported that concanavalin A was superior to PHA or anti-CD3 (OKT3; as utilised in our study protocol) as a γδ T cell mitogen. Additionally, γδ T cell cytotoxicity may be disease-stage dependent, as a recent study has demonstrated that expanded γδ T cells display reduced cytotoxic potential against increasing CLL cell burden [53].

The γδ T cells used in the present study contained variable numbers of Vδ1, Vδ2 or Vδ3 T cells and it is possible that each population can influence each other when in culture. Data from Siegers et al. [25] suggests that activated Vδ2 T cells can inhibit Vδ1 T cells. Therefore, it may be worth selecting for and expanding Vδ1 T cell lines, rather than using

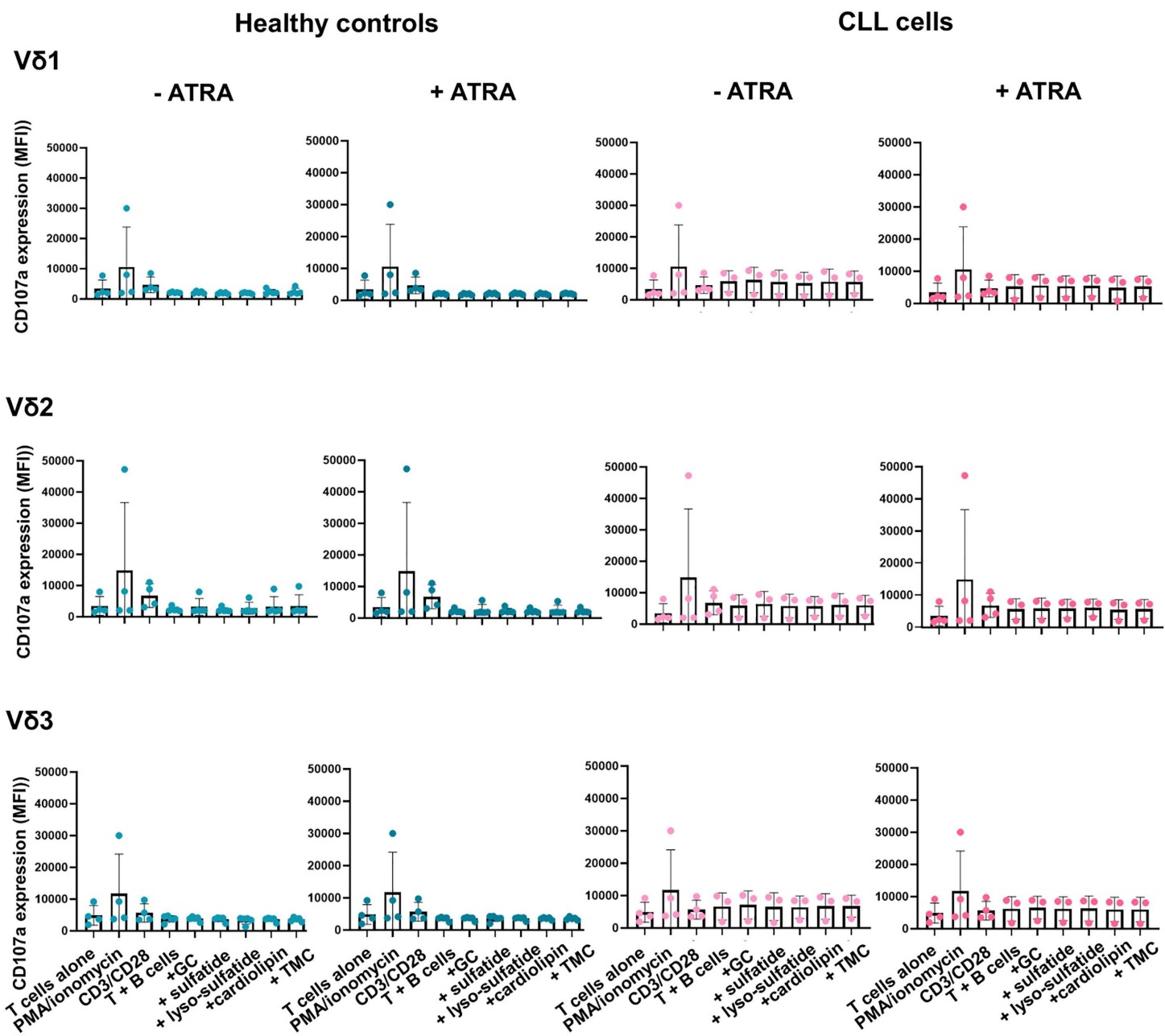

**Fig 7. Treatment of 100 ng/mL ATRA in addition of 5 different glycolipids; α-galactosylceramide (GC), sulfatide, lyso-sulfatide, cardiolipin and TMC to B cells from healthy donors and CLL patients did not sensitise them for cytolytic degranulation by γδ T cells.** Enriched B cells from 5 healthy donors and PBMCs from 4 CLL patients were treated with or without 100 ng/mL ATRA for 72 hours and 5 different glycolipids were sonicated and added and then incubated for an additional 24 hours. The cells were then co-cultured with expanded γδ T cell lines from healthy donors in the presence of an anti-CD107a mAb (n = 5). Cells were then stained with mAbs specific for CD3, CD19, Vδ1, Vδ2 and Vδ3 and analysed by flow cytometry. Scatter bar plot indicates the mean (±SD) fluorescence intensities (MFI) of CD107a expression by gated Vδ1 (top row), Vδ2 (centre row), and Vδ3 (bottom row) T-cells. A negative control (T cells alone) and positive control (PMA & Ionomycin, anti-CD3/anti-CD28) were included (n = 4). MFIs were compared using One-way ANOVA and post hoc Dunnett's multiple comparison test and no significant differences were found.

polyclonal pan γδ lines. Another difference between our γδ T cell lines and those used by Poggi et al. [24], is that theirs' were derived from CLL patients and not healthy donors. Lines generated from patients with CLL have likely been exposed to circulating CLL cells *in vivo*, which may influence their cytotoxic potential or re-invoke a memory response upon secondary exposure to antigens [54,55]. The E:T ratios of 20:1, 10: and 5:1 used in other studies [24,25,27] also differ from the E:T ratio of 1:1 used in our study and may explain why no significant killing was observed. Moreover, functional exhaustion of long-term expanded γδ T cell lines, as seen by moderate expression of PD-1, CTLA-4, and TIM-3 on Vδ1 and Vδ3 subsets, may in part contribute to the lack of cytotoxicity observed against CLL cells in this study.

Our study was limited by a small sample size and highly variable patient cohort. Future studies should include larger, stratified cohorts, and compare γδ T cells expanded using different methods. The use of purified Vδ1, Vδ2 or Vδ3 T cell cultures may prove more effective in the killing of primary CLL cells.

In conclusion, this study confirms that both Vδ1 and Vδ3 T cells are significantly expanded in the blood of CLL patients, and that CLL B cells downregulate key ligands (CD1d and ULBP3) required for γδ T cell activation. Although treatment with RARα agonists like ATRA and AM580 can restore these ligand levels, this alone does not render CLL cells susceptible to γδ T cell killing. Similarly, pulsing with synthetic glycolipids failed to activate cytotoxicity. These findings challenge the idea that RAR-α agonists or glycolipid-based priming can sufficiently activate γδ T cells against CLL and underscore the complexity of immune evasion in CLL. Nevertheless, the study highlights the potential of γδ T cells as immunotherapeutic agents, provided that optimal activation conditions and cytotoxic subsets can be identified. Despite not being a dominant endogenous anti-CLL mechanism, γδ T cell cytotoxicity warrants further investigation, given the resistance associated with anti-CD20 therapies and the advantage of being able to use γδ T cells as an allogeneic therapy.

## Supporting information

**S1 Fig. Flow cytometric analysis of B cells and γδ T cell subsets.**
(TIF)

**S2 Fig. Identification of CD1d-binding glycolipids with agonist T cell activity within human PBMC and purified Vδ1 T cells.**
(TIF)

**S1 Table. David et al.Raw data.**
(XLSX)

## Acknowledgments

The authors are grateful to the donors who participated in this study. We are indebted to the Irish Blood Transfusion Service for providing buffy coat packs as a source of cells for this study.

## Author contributions

**Conceptualization:** Anthony M. McElligott, Derek G. Doherty.

**Data curation:** Julie David, Derek G. Doherty.

**Formal analysis:** Julie David.

**Investigation:** Amy Walsh, Ke Sin Seow, Stefan Elekes, Rohit Upadhyay, Nawal A. B. Taher, Samiha Al Siyabi, Ashanty M. Melo, Carmel Waldron, Anthony M. McElligott.

**Methodology:** Julie David, Amy Walsh, Ke Sin Seow, Ellen Walsh, Stefan Elekes, Rohit Upadhyay, Nawal A. B. Taher, Samiha Al Siyabi, Ashanty M. Melo, Elizabeth Vandenberghe, Anthony M. McElligott, Derek G. Doherty.

**Project administration:** Derek G. Doherty.

**Resources:** Ellen Walsh, Carmel Waldron, Elizabeth Vandenberghe, Anthony M. McElligott.

**Supervision:** Anthony M. McElligott, Derek G. Doherty.

**Writing – original draft:** Julie David, Derek G. Doherty.

**Writing – review & editing:** Julie David, Amy Walsh, Ke Sin Seow, Ellen Walsh, Stefan Elekes, Rohit Upadhyay, Nawal A. B. Taher, Samiha Al Siyabi, Ashanty M. Melo, Carmel Waldron, Elizabeth Vandenberghe, Anthony M. McElligott, Derek G. Doherty.

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
