## [Decision Letter · Decision Letter 0]

9 Dec 2025

Dear Dr. Doherty,

Thank you for submitting your manuscript to PLOS ONE. After careful consideration, we feel that it has merit but does not fully meet PLOS ONE’s publication criteria as it currently stands. Therefore, we invite you to submit a revised version of the manuscript that addresses the points raised during the review process.

We look forward to receiving your revised manuscript.

Kind regards,

Xiaosheng Tan

Academic Editor

PLOS One

**Journal Requirements:**

4. . Please upload a new copy of Figures 1, 2, 3, 4, 5, 6, 7; and Supplementary Figure 1  as the detail is not clear. Please follow the link for more information:  https://journals.plos.org/plosone/s/figures

5. We note that there is identifying data in the Supporting Information file <David et al. Raw data.xlsx>. Due to the inclusion of these potentially identifying data, we have removed this file from your file inventory. Prior to sharing human research participant data, authors should consult with an ethics committee to ensure data are shared in accordance with participant consent and all applicable local laws.

-Location data

Additional guidance on preparing raw data for publication can be found in our Data Policy (httpps://journals.plos.org/plosone/s/data-availability#loc-human-research-participant-data-and-other-sensitive-data) and in the following article: http://www.bmj.com/content/340/bmj.c181.long.

**Additional Editor Comments:**

Please respond to reviewers' comments individually.

Reviewers' comments:

Reviewer's Responses to Questions

**Comments to the Author**

1. Is the manuscript technically sound, and do the data support the conclusions?

Reviewer #1: Yes

Reviewer #2: Partly

2. Has the statistical analysis been performed appropriately and rigorously?

Reviewer #1: Yes

Reviewer #2: Yes

3. Have the authors made all data underlying the findings in their manuscript fully available?

Reviewer #1: Yes

Reviewer #2: Yes

4. Is the manuscript presented in an intelligible fashion and written in standard English?

Reviewer #1: Yes

Reviewer #2: Yes

Reviewer #1: Julie et al. present a study that provides valuable insights into the regulation of CD1d and ULBP3 on B cells from healthy donors and CLL patients and their inability to sensitize these cells to γδ T-cell–mediated killing. While the findings are clearly presented, addressing the following points would substantially strengthen the study’s scientific rigor, clarify mechanistic causality, and enhance its translational relevance.

Major Comments:

1. The cytotoxicity assays depend almost exclusively on CD107a surface expression as a surrogate for γδ T-cell degranulation. Although CD107a is a widely used marker, it does not always correlate with actual target-cell death, particularly in cases where immunological synapse formation or perforin-mediated killing is impaired. Incorporating direct cytotoxicity measurements—such as flow-based live/dead killing assays, or real-time imaging—across multiple effector-to-target ratios and time points would provide more definitive evidence regarding whether B cells are truly resistant to γδ T-cell–mediated killing.

2. All functional assays rely on long-term expanded γδ T-cell lines derived from healthy donors. Such expanded lines may exhibit altered receptor expression or activation thresholds, and allogeneic interactions do not accurately model autologous physiology. Including freshly isolated T cells and ideally autologous γδ T cells from the same individuals would help exclude artifacts arising from in vitro expansion or allogeneic mismatch and strengthen confidence in the negative findings.

3. RAR-α agonist treatment may induce additional immunomodulatory molecules on B cells beyond CD1d and ULBP3. Assessing inhibitory ligands such as PD-L1, or other immunosuppressive markers would clarify whether these pathways contribute to the lack of γδ activation. Furthermore, the study does not evaluate whether RAR-α agonists directly alter γδ T-cell viability, metabolic state, or cytokine production/proliferation. These controls are important to exclude unintended suppression or modulation of γδ T-cell function by the compounds.

4. Additional functional profiling would provide a more complete understanding of γδ T-cell activation status. Quantifying granzyme B, perforin, IFN-γ, TNFα, and activation markers such as CD69 or CD25 would help determine whether γδ T cells exhibit partial, non-cytolytic, or exhausted activation states not reflected by CD107a alone.

Reviewer #2: The manuscript reports CLL patients have reduced B-cell CD1d and increased Vδ1/Vδ3 frequencies and aims to test whether restoration of CD1d expression on B cells can sensitize CLL cells to γδ T-cell-mediated cytotoxicity. While this is an interesting and clinically relevant question, the current experimental evidence is not sufficient to conclude that γδ T cells fail to respond to CD1d-restored B cells. Further experiments and clarifications could strengthen the results. Please see my comments below.

Abstract

Authors claim that “This study reveals that human Vδ1 T cell cytotoxicity against CLL B cells may be disease stage dependent and requires B cell priming and selective expansion and activation of specific γδ T cell subsets.” However, no results or discussion are presented regarding how cytotoxicity varies with disease stage.

Introduction:

The introduction is clear and provides sufficient background to set the context and the relevance of this study.

Methods:

The authors report using an E:T ratio of 1:1 in their co-cultures. While this is acceptable, γδ T-cell-mediated responses often require higher ratios (e.g., 5:1 or 10:1) to detect degranulation and target killing, particularly against primary B cells.

Results:

While CD107a reflects degranulation, it does not necessarily demonstrate that target cells are being killed. Complement CD107a with direct granzyme B delivery and target death are recommended.

Besides the E:T ratio and complementary assays for CD107a, the negative results may also reflect (a) the need of subset purification/priming, Vδ1 T cells ranged from 13.9 to 51%, which means in some cases, Vδ1 could be too diluted to show changes. Vδ3 T cells are a minor population (1.3-25%), so CD107a negativity in this subset could be due to low numbers rather than true absence of degranulation. (b) the possibility that Vδ1/Vδ3-mediated cytotoxicity proceeds via non-degranulation pathways, which would not be captured by CD107a.

Line 308 states PBMC from 13 CLL patients were analyzed but Table 1 indicates there are 15 patients. Please explain.

Figure 3F, why is sample number in ULBP3 detection different from the sample number in CD1d figure?

Figure 5 does not show ULBP3 expression after treatment in CLL donors.

Figure 6 showed healthy donor T cells alone show even higher CD107a than the positive control (PMA/ionomycin), which requires explanation.

Discussion:

Discussion comparing these results to previous work with iNKT-mediated killing would help interpret the negative findings.

Since γδ-mediated killing is not a major mechanism in CLL, providing rationale for studying this mechanism, e.g., limitations of anti-CD20 therapies and potential additive effects of alternative cytotoxic pathways, would strengthen the significance.

Figures:

Gating strategy figure (Figure 1) should be moved to supplementary materials.

Figure 5, please keep colors consistent with other figures.

Figure 7, please label for Vδ1/2/3 T cells in figures to improve readability.

Figure legends appear misplaced.

.

Reviewer #1: **Yes:** Jing JuJing JuJing JuJing Ju

Reviewer #2: No

---

## [Author Response · Author response to Decision Letter 1]

10 Mar 2026

Responses to all reviewer and editor comments have been uploaded in the file entitled 'Response to reviewers'

---

## [Decision Letter · Decision Letter 1]

23 Mar 2026

Upregulation of CD1d and ULBP3 on B cells from healthy donors and chronic lymphocytic leukaemia patients does not prime them for killing by γδ T cells

PONE-D-25-60204R1

Dear Dr. Doherty,

We’re pleased to inform you that your manuscript has been judged scientifically suitable for publication and will be formally accepted for publication once it meets all outstanding technical requirements.

Kind regards,

Xiaosheng Tan

Academic Editor

PLOS One

Additional Editor Comments (optional):

Reviewers' comments:

Reviewer's Responses to Questions

**Comments to the Author**

Reviewer #1: (No Response)

Reviewer #2: All comments have been addressed

2. Is the manuscript technically sound, and do the data support the conclusions?

Reviewer #1: (No Response)

Reviewer #2: Yes

3. Has the statistical analysis been performed appropriately and rigorously?

Reviewer #1: Yes

Reviewer #2: Yes

4. Have the authors made all data underlying the findings in their manuscript fully available?

Reviewer #1: Yes

Reviewer #2: Yes

5. Is the manuscript presented in an intelligible fashion and written in standard English?

Reviewer #1: Yes

Reviewer #2: Yes

Reviewer #1: While a few comments remain partially unresolved, most of the major concerns have been properly addressed. In my opinion, the revised manuscript meets the standards for acceptance.

Reviewer #2: I appreciate the authors’ detailed responses, revisions, and additional experiments, which have strengthened the manuscript and clarified several points. I believe the manuscript is now suitable for publication.

.

Reviewer #1: No

Reviewer #2: No

---

## [Editor Report · Acceptance letter]

PONE-D-25-60204R1

PLOS One

Dear Dr. Doherty,

I'm pleased to inform you that your manuscript has been deemed suitable for publication in PLOS One. Congratulations! Your manuscript is now being handed over to our production team.

Kind regards,

on behalf of

Dr. Xiaosheng Tan

Academic Editor

PLOS One